# ARE UNIFIED VISION-LANGUAGE MODELS NECESSARY: GENERALIZATION ACROSS UNDERSTANDING AND GENERATION

## ABSTRACT

Recent advancements in unified vision-language models (VLMs), which integrate both visual understanding and generation capabilities, have attracted significant attention. The underlying hypothesis is that a unified architecture with mixed training on both understanding and generation tasks can enable mutual enhancement between understanding and generation. However, this hypothesis remains underexplored in prior works on unified VLMs. To address this gap, this paper systematically investigates the generalization across understanding and generation tasks in unified VLMs. Specifically, we design multiple datasets closely aligned with real-world scenarios to facilitate extensive experiments and quantitative evaluations. We evaluate multiple unified VLM architectures to validate our findings. Our key findings are as follows. **First**, unified VLMs trained with mixed data exhibit mutual benefits in understanding and generation tasks across various architectures, and this mutual benefits can scale up with increased data. **Second**, alignment between multimodal input and output spaces is important to mutual benefits. Better alignment will lead to more significant mutual benefits. **Third**, the knowledge acquired during generation tasks can transfer to understanding tasks, and this cross-task generalization occurs within the base language model, beyond modality adapters. Our findings underscore the critical necessity of unifying understanding and generation in VLMs, offering valuable insights for the design and optimization of unified VLMs.

## 1 INTRODUCTION

In recent years, Vision-Language Models(VLMs) has emerged as a transformative paradigm in artificial intelligence. These models are typically categorized into two distinct types: *understanding-only VLMs* (Chen et al., 2024b; Liu et al., 2023; Wang et al., 2024a), which focus mainly on comprehension and perception tasks like visual question answering (VQA) and image captioning; and *generation-only VLMs* (Betker et al., 2023; Podell et al., 2023; Tian et al., 2024b), which excel in tasks like image generation and image editing. Although these specialized models have achieved remarkable success in their respective domains, recent research has increasingly shifted toward the development of *unified VLMs* (Chen et al., 2025; Xie et al., 2024; Zhou et al., 2024; Wang et al., 2024b; Team, 2024). These unified models aim to integrate both understanding and generation capabilities within a single framework.

The intuitive motivation for developing unified VLMs stems from the hypothesis that a shared architecture and mixed training across understanding and generation tasks can foster mutual benefits. As Richard Feynman famously stated, "What I cannot create, I do not understand." This philosophy underscores the potential synergy between understanding and generation. Specifically, it is hypothesized that the knowledge acquired through understanding tasks can be leveraged to enhance performance on generation tasks. For instance, spatial concepts learned during image caption tasks may assist the model in generating images correctly following complex text instructions. Conversely, successful execution of generation tasks may hinge on the model's ability to comprehend the underlying concepts in textual instructions. This process, in turn, can reinforce the model's understanding of these concepts. For example, generating images with precise spatial relationships may deepen the model's grasp of spatial concepts, thereby improving its performance on related understanding tasks.

Despite the growing interest in unified VLMs, most existing works have predominantly focused on architectural innovations or training strategies. However, a critical question remains largely unexplored: Are unified vision-language models (VLMs) truly necessary when separate models for understanding and generation already excel in their respective domains? Currently, unified VLMs do not exhibit clear superiority over separate models in either understanding or generation tasks(Chen et al., 2025; Team, 2024). Moreover, the generalization across understanding and generation tasks has been insufficiently studied. This topic has only been briefly discussed in a few prior works (Tong et al., 2024; Chen et al., 2025; Wu et al., 2024a), often as a secondary consideration within the context of specific architectures. It remains unclear whether such generalization consistently exists across different unified VLMs and what factors influence this phenomenon.

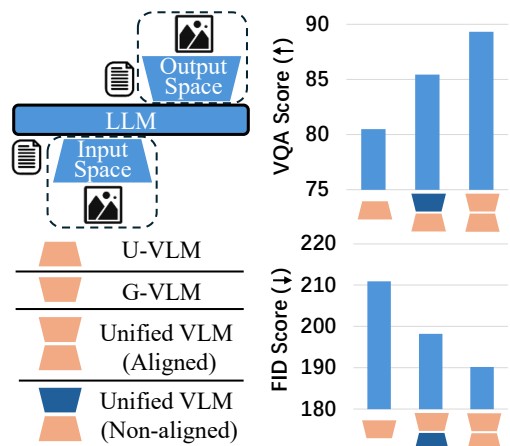

Figure 1: Unified VLMs surpass understanding-only and generation-only models. Alignment of vision input-output spaces further boosts performance. Results from Section4.

To address this gap, this paper systematically investigates the generalization across understanding and generation in unified VLMs. First, to facilitate extensive experimentation, we carefully design two easy-to-control image-text dataset aligned with real-world scenarios. This dataset includes both vision understanding data (e.g., visual question answering (VQA) and image captioning) and vision generation data (e.g., text-to-image generation). We then evaluate and analyze multiple unified VLM architectures, covering a wide range of prior works. Our key findings are as follows:

*First, unified VLMs exhibit mutual benefits between understanding and generation tasks.* Specifically, we compare the performance of unified VLMs trained with mixed tasks (combining understanding and generation) against task-specific models trained solely on either understanding or generation tasks using the same set of data. The results show that most of the unified VLMs outperform their task-specific counterparts, as illustrated in Figure 1. To further explore these mutual benefits, we conduct experiments by fixing the amount of understanding data while increasing the generation data, and vice versa. We find that the mutual benefits can scale up with the increase in training data, highlighting the synergy between understanding and generation.

*Second, better alignment between vision input and output spaces leads to improved generalization.* Our evaluations reveal that unified VLMs with well-aligned vision input and output spaces exhibit more pronounced mutual benefits, as shown in Figure 1. To validate this observation, we introduce artificial distortions to disrupt the alignment between the vision input and output spaces. The results demonstrate that such disruptions significantly reduce the mutual benefits in unified VLMs, while task-specific models remain largely unaffected. These findings underscore the critical role of alignment between vision input and output spaces in facilitating cross-task generalization.

*Third, knowledge acquired during generation tasks can transfer to understanding tasks.* Leveraging our carefully designed synthetic dataset, we simulate scenarios where specific knowledge is underrepresented in the understanding data but remains present in the generation data. While understanding-only models struggle to learn this knowledge, unified VLMs successfully acquire it and achieve near-perfect accuracy on related tasks. This empirically demonstrates the transfer of knowledge from generation tasks to understanding tasks. Further analysis reveals that this knowledge transfer occurs primarily within the base language model (LLM), beyond the modality adapters

Through these findings, we highlight the necessity of unifying understanding and generation within a single framework, as evidenced by the mutual benefits observed in unified VLMs. Additionally, we reveal the potential for scaling up vision-language models through the integration of understanding and generation tasks. We hope that our experimental pipeline will facilitate future research in this area and provide insights for the design and optimization of unified VLMs.

## 2 RELATED WORK

**Unified Vision-Language Models.** Recent efforts have aimed to build unified VLMs that seamlessly support both understanding and generation across vision and language modalities. A popular direction extends autoregressive language modeling to both text and image tokens (Wu et al., 2024b; Wang et al., 2024b), enabling a single transformer to predict the next token regardless of modality. Models such as LWM (Liu et al., 2024) and Chameleon (Team, 2024) adopt discrete VQ-based image tokenizers (Gafni et al., 2022), allowing vision-language inputs to be encoded and decoded within a unified autoregressive framework. Beyond token-based approaches, diffusion models have recently emerged as a powerful tool for vision generation. A group of works treats diffusion as an external module: an autoregressive LLM first generates latent codes, which are then passed to a pretrained diffusion model to produce the final image (Dong et al., 2023; Tian et al., 2024a). Transfusion (Zhou et al., 2024) and Showo (Xie et al., 2024) take a hybrid route, integrating continuous or discrete diffusion for images with autoregressive text prediction, offering greater flexibility for mixed-modal generation. Alternative strategies explore architectural innovations to support dual capabilities. Janus-pro (Chen et al., 2025) decouples vision encoders into separate pathways to balance understanding and generation tasks. Dual Diffusion (Li et al., 2024) proposes using two independent diffusion processes for the two capabilities, while Liquid (Wu et al., 2024a) aligns visual and textual representations in a shared space for unified token-level modeling.

**Generalization Across Generation and Understanding.** While unified MLLMs have demonstrated promising performance in both understanding and generation tasks, their purported advantage of mutual enhancement across modalities remains an open question. Existing studies have largely focused on model architecture or training efficiency without systematically evaluating the generalization capability across understanding and generation paradigms (Team, 2024; Sun et al., 2023; Wang et al., 2024b; Chen et al., 2025). Although some recent works have begun to explore this intersection, they reach inconsistent conclusions, highlighting critical gaps. Tong et al. (2024) observe consistent mutual benefits in a unified VLM that uses SigLIP (Zhai et al., 2023) as the vision encoder for understanding tasks and generates SigLIP tokens, which are used as conditions for a diffusion model to generate images. They also find that these mutual benefits scale with increased data. In contrast, Wu et al. (2024a) discuss the impairment of both understanding and generation tasks in a unified VLM that uses VQ-VAE (Van Den Oord et al., 2017) as the vision encoder for understanding tasks and generates VQ-VAE tokens. Their experiments reveal that this impairment diminishes as the model size increases. Similarly, Chen et al. (2025) claim that architectural disentanglement of vision understanding and generation can alleviate conflicts between these tasks. Using SigLIP as the vision encoder and generating VQ-VAE tokens for image generation tasks, their experiments reveal inconsistent mutual effects: in some tasks, understanding and generation benefit each other, while in others, they harm one another. Despite these initial efforts, the above analyses are restricted to specific unified VLM architectures, and none provide a comprehensive evaluation of bidirectional generalization across understanding and generation tasks across different unified VLM designs. Moreover, these studies are computationally intensive and difficult to reproduce. Our work addresses this gap by offering a systematic study across various unified VLMs, leveraging carefully curated datasets and finely controlled, computationally friendly experiments.

## 3 PRELIMINARIES

### 3.1 DATASETS

To ensure precise control over the dataset, facilitate flexible adjustments to its distribution for analysis, and reduce computational cost for numerous experiments, we mainly use two datasets to conduct our experiments: a synthetic SmartWatch dataset and a modified CelebA (Liu et al., 2015) dataset. We use a combination of 60K VQA data, 60K caption data and 60k text-to-image generation data as the default training dataset.

**SmartWatch.** This dataset closely mimics real-world scenarios for both image understanding and generation tasks, as illustrated in Figure 2. Each image in the dataset is controlled by six distinct attributes: time, weather, weather position, battery level, battery position, and watch face color. Specifically: Time consists of hour, minute, and second values ranging from 0–12, 0–60, and 0–60, respectively. Weather can take on three states: cloudy, rainy, or sunny. Battery level ranges from 0 to

100. Both weather position and battery position can be top-left, top-right, bottom-left, or bottom-right. To generate each data sample, we first define the ground truth for the six attributes. A rule-based generator then creates the corresponding image based on these ground truth attributes, ensuring accurate pairing between image and text data. The watch face color is randomly sampled and not included in any text data, allowing a single text ground truth to correspond to multiple images. This design simulates real-world scenarios where diverse visual variations may arise from the same textual description, as shown in Figures 2a. For understanding tasks, we generate VQA (Visual Question Answering) and caption data using diverse QA templates. Each VQA question focuses on one attribute, with equal appearance probabilities for the five attributes. For caption data, each caption includes 1–5 attributes, with time always present and the other four appearing independently with a probability of 0.5. For generation tasks, we create instructions following the caption data using different templates.

**CelebA.** We select seven clearly defined attributes in the original CelebA: 'Black_Hair', 'Eyeglasses', 'No_Beard', 'Male', 'Wearing_Hat', 'Wearing_Necklace', 'Wearing_Necktie'. To generate one image-text pair, we first randomly sample to obtain the ground truth of the attributes, then generate the text part using pre-defined templates. After that we filter and randomly sample the corresponding image. Images for training and testing are from two non-overlapping image pools respectively. The procedure to generate VQA data, caption data and image generation data is the same as SmartWatch.

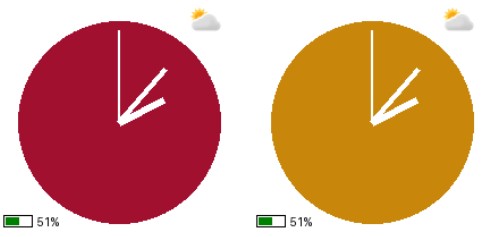 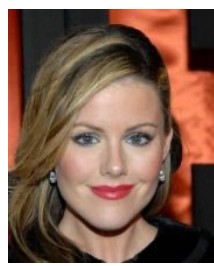

(a) Samples from the SmartWatch dataset showing time 02:07:00, cloudy weather displayed at the top-right, and 51% battery displayed at the bottom-left.

(b) Samples from the CelebA dataset showing person who does not have black hair, has no beard, is a female, is not wearing a hat, is not wearing a necklace.

Figure 2: Samples from SmartWatch and CelebA with the corresponding ground truth attributes.

### 3.2 EVALUATION

The evaluation of understanding tasks is conducted in the VQA format. For generation tasks, we compute the FID score (Heusel et al., 2017) between the ground truth images and the generated ones.

### 3.3 UNIFIED VLMS

In this paper, we primarily focus on LLM-based Unified Vision-Language Models (VLMs). For image understanding, a pre-trained image encoder first encodes the input image. Subsequently, a dedicated understanding vision adapter projects the encoded representations from the encoder's hidden space to the LLM's hidden space, transforming them into input vision tokens. These vision tokens are then combined with text tokens and fed into the LLM. The pre-trained image encoder can be either a VQ-VAE encoder(Van Den Oord et al., 2017) or a SigLIP vision encoder (Zhai et al., 2023). For image generation, the LLM generates a special symbol, "¡image¿", at the beginning of the generation process. Upon encountering this symbol, an image generation head is activated instead of the language modeling head. This image generation head projects the LLM's output hidden states into vision tokens. After generating the first vision token, a generation vision adapter maps these tokens back to the LLM's input hidden space to enable autoregressive generation. The parameters of the generation vision adapter can optionally be shared with those of the understanding vision adapter, depending on whether the input and generated vision tokens reside in the same latent space. The generated vision tokens may correspond to SigLIP vision embeddings or VQ-VAE tokens.

## 3.4 EXPERIMENT SETTINGS

We evaluate various combinations of VQ-VAE and SigLIP within unified VLMs:

**SigLIP-VQ**: SigLIP serves as the vision encoder for the VLM, generating VQ token IDs that are decoded into real images by the VQ-VAE decoder, align with Janus Chen et al. (2025).

**VQ-VQ**: The VQ-VAE encoder acts as the vision encoder for the VLM, generating VQ token IDs that are decoded into real images by the VQ-VAE decoder, resembling Liquid Wu et al. (2024a).

**SigLIP-SigLIP**: SigLIP is used as the vision encoder for the VLM, generating SigLIP embeddings align with MetaMorph Tong et al. (2024).

**VQ-SigLIP**: The VQ-VAE encoder serves as the vision encoder for the VLM, generating SigLIP embeddings. While this configuration is not practical for real-world applications, it serves as a valuable baseline for comparison in our experiments.

For VQ-VAE, we use `vq_ds16_t2i` from Sun et al. (2024), with a resolution of $256 \times 256$. For SigLIP, we use `SigLIP-base-patch16-224` from Zhai et al. (2023), with a resolution of $224 \times 224$. In unified VLMs with aligned input and output vision spaces, we share the parameters of the understanding vision adapter and generation vision adapter by default to maintain alignment. For the base LLM in all unified VLMs, we use `Vicuna-7B-v1.5` (Peng et al., 2023) for fair comparison. We adopt a one-stage training approach to jointly update the vision adapters, image generation head, and LLMs. By default, we fine-tune the base LLM using Low-Rank Adaptation (LoRA). More details are in the supplementary materials.

# 4 GENERALIZATION ACROSS UNDERSTANDING AND GENERATION

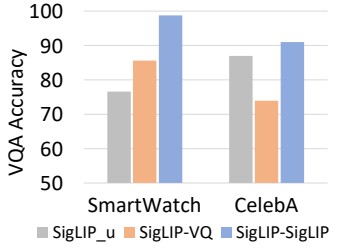
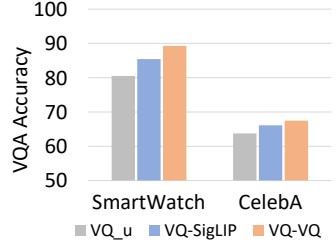
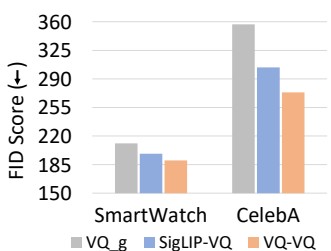

(a) Image understanding performance of SigLIP-as-vision-encoder unified VLM.

(b) Image understanding performance accuracy of VQVAE-as-vision-encoder unified VLMs.

(c) Image generation performance of VQVAE-as-vision-decoder unified VLMs.

Figure 3: Image understanding and generation performance of VLMs during training. "_g" denotes generation-only training, and "_u" denotes understanding-only training. Most Unified VLMs trained with mixture of understanding and generation data outperform task-specific models trained with understanding-only or generation-only data. Unified VLMs with aligned vision input and output space (SigLIP-SigLIP and VQ-VQ) performs better.

First, we train the four types of unified VLMs using a mixture of understanding and generation data. To verify whether unified VLMs can yield mutual benefits between understanding and generation compared to task-specific models, we compare their performance with models trained solely on understanding-only or generation-only data. The evaluation results are shown in Figure 3. From the figure, we observe that most unified VLMs trained with mixed data outperform their task-specific counterparts. For understanding tasks, SigLIP-SigLIP surpass SigLIP_u, SigLIP-VQ surpass SigLIP_u on SmartWatch, while VQ-VQ and VQ-SigLIP outperform VQ_u. For generation tasks, both SigLIP-VQ and VQ-VQ exceed the performance of VQ_g. These results indicate that generalization across understanding and generation tasks does indeed exist in unified VLMs. Training on both tasks can enhance the performance of each other, empirically demonstrating the necessity and superiority of unified VLMs over task-specific models.

From Figure 3, we observe that unified VLMs with aligned vision input and output spaces tend to perform better. Specifically, SigLIP-SigLIP outperforms SigLIP-VQ in understanding tasks, VQ-VQ

Table 1: Effect of the affine transformation on model performance. SW and CA denote the Smart-Watch and CelebA datasets, respectively. Values in parentheses indicate the performance change after applying the transformation. The transformation has a negligible impact on non-unified models but significantly degrades the performance of unified models.

| Model Type | Transform | SigLIP_u | | VQ_u | | VQ_g | |
|---|---|---|---|---|---|---|---|
| | | SW VQA ↑ | CA VQA ↑ | SW VQA ↑ | CA VQA ↑ | SW FID ↓ | CA FID ↓ |
| Non-Unified | w\o-Affine | 76.6 | 87.0 | 80.5 | 63.8 | 210.9 | 357.0 |
| | w-Affine | 78.2(+1.6) | 87.3(+0.3) | 78.9(-1.6) | 63.3(-0.5) | 210.9(-0.0) | 357.0(-0.0) |

| Model Type | Transform | SigLIP-SigLIP | | VQ-VQ | | | |
|---|---|---|---|---|---|---|---|
| | | SW VQA ↑ | CA VQA ↑ | SW VQA ↑ | CA VQA ↑ | SW FID ↓ | CA FID ↓ |
| Unified | w\o-Affine | 98.7 | 91.0 | 89.3 | 66.1 | 190.2 | 273.5 |
| | w-Affine | 91.3(-7.5) | 85.3(-5.8) | 85.1(-4.2) | 64.5(-1.6) | 195.8(+5.6) | 368.4(+94.9) |

surpasses VQ-SigLIP in understanding tasks, and VQ-VQ outperforms SigLIP-VQ in generation tasks. This discrepancy motivates us to hypothesize the generalization of knowledge between generation and understanding is influenced by the distance between the vision input and output spaces.

To further validate this hypothesis, we introduce a random affine transformation immediately after the understanding vision adapter to distort the vision input space, making it slightly different from the vision output space. We carefully ensure that the affine transformation is reversible to avoid information loss. The results are shown in Table 1. First, the affine transformation has little effect on the performance of understanding-only VLMs. This guarantees that any performance difference in unified VLMs with or without the affine transformation arises from the misalignment of the input and output vision spaces, rather than from the distortion of the input space itself. Based on this, mixed training with the affine transformation performs worse than without the transformation in both understanding and generation tasks for both SigLIP-SigLIP and VQ-VQ. The performance degradation is significant, even leading to the underperformance of unified models compared to non-unified models. This empirically confirming our hypothesis: Understanding can benefit from generation, but this benefit depends on the alignment between the vision input and output spaces. This may be because, when the input and output spaces are close or identical, the embeddings of the same visual concept are similar across understanding (input) and generation (output), facilitating easier learning. Conversely, when the two spaces are distant, it becomes challenging for the base LLM to capture the relationship between input and output representations of the same visual concept, leading to poor generalization, even causing conflicts between understanding and generation. This also explains the underperformance of SigLIP-VQ compared to SigLIP_u in Figure 3.

## 4.1 SCALING UP WITH INCREASED DATA

Building on the previous findings, we investigate whether the mutual benefits between understanding and generation can scale with increased training data. To this end, we expand the dataset in two directions: (1) fixing the generation data at 60K while increasing the understanding data from 0K to 120K, 180K, 240K, and 300K; and (2) fixing the understanding data at 120K while increasing the generation data from 0K to 60K, 90K, 120K, and 180K. The evaluation results are shown in Figure 4.

Our results reveal two key trends regarding the interplay between understanding and generation data. First, as shown in the right panel of Figure 4a, when we increase the amount of understanding data while holding generation data constant, the generation performance (measured by FID score) exhibits one of two patterns: it either improves continuously or improves initially before declining. Conversely, a similar trend is observed for understanding performance (VQA accuracy) when increasing the amount of generation data while keeping understanding data fixed, as depicted in the left panel of Figure 4b. A notable exception is SigLIP-VQ on the CelebA dataset, where performance degrades with 60K generation samples, which can likely be attributed to the aforementioned misalignment between the vision input and output spaces. These findings have two significant implications:

First, the initial performance boost in one task from adding data for the other validates the strong cross-task generalization of unified VLMs. This synergy suggests that generation training can enhance understanding, and vice versa, reinforcing the value of a unified modeling approach.

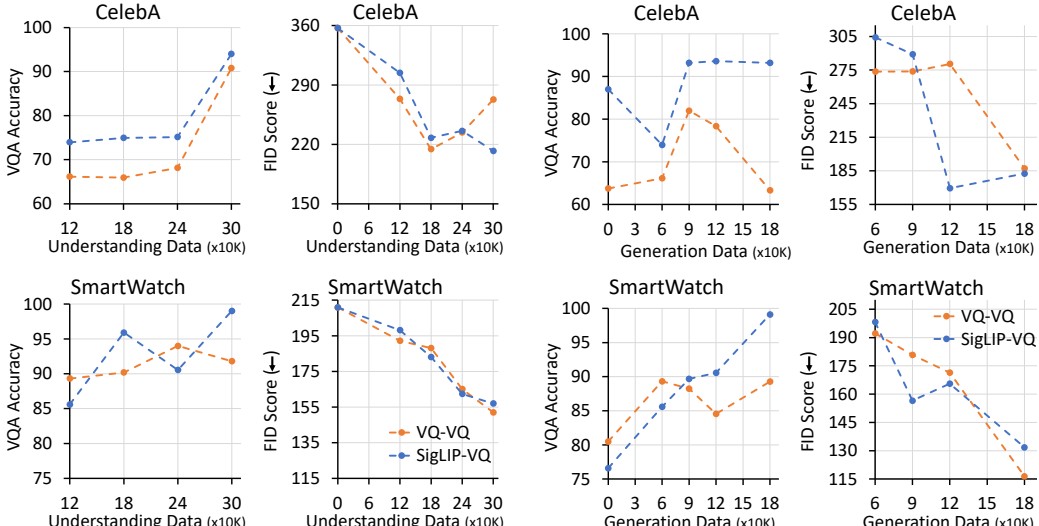

(a) Performance of understanding and generation as the amount of understanding data increases, with generation data fixed at 60K.

(b) Performance of understanding and generation as the amount of generation data increases, with understanding data fixed at 120K.

Figure 4: Performance of SigLIP-VQ and VQ-VQ under varying data scales. Only increase the amount of generation data can boost the performance in understanding tasks, and vice versa.

Second, The eventual performance decline underscores the critical need for balance between the amounts of understanding and generation data. Once one data type becomes dominant in the training mixture, further increasing its volume can harm performance on the complementary task, leading to a performance trade-off instead of mutual reinforcement.

## 4.2 KNOWLEDGE TRANSFER FROM GENERATION TO UNDERSTANDING

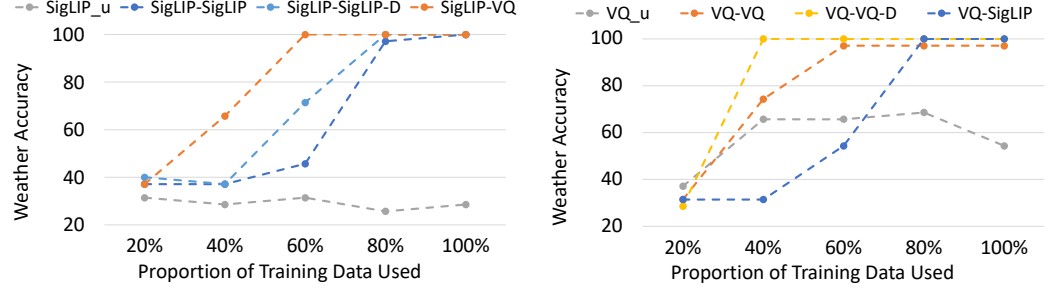

Figure 5: Performance comparison between unified VLMs and understanding-only VLMs, trained on understanding data biased on the weather attribute. "_u" denotes understanding-only training. "-D" denotes using separate understanding vision adapter and generation vision adapter.

Based on the previous findings that adding generation data can improve understanding performance (and vice versa), we hypothesize that knowledge learned in one task (e.g., generation) can transfer to the other task (e.g., understanding) in unified VLMs. Leveraging our easy-to-control synthetic dataset, we specifically manipulate the appearance of key attributes to test this hypothesis. In this experiment, we intentionally reduce the occurrence of one attribute in understanding tasks to a very low level (0 appearances in image captioning data and 0.05 probability of appearance in VQA data), while maintaining normal generation data. We apply this manipulation to two attributes, weather and battery, creating two biased datasets: one biased on weather and the other biased on battery. We then evaluate and compare the performance of understanding-only VLMs with that of unified VLMs. The results are shown in Figures 5 and 6.

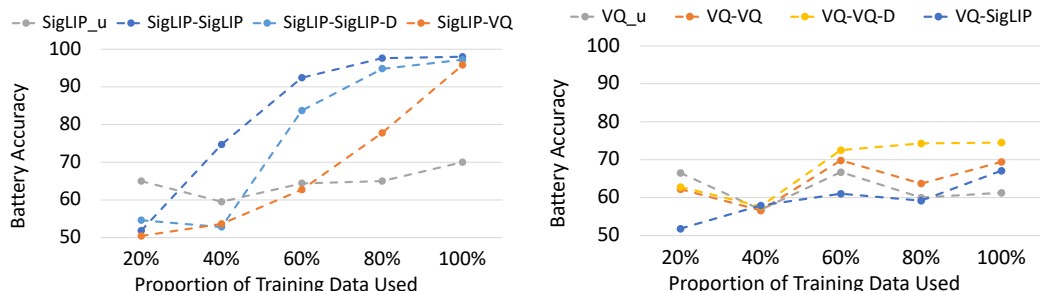

Figure 6: Performance comparison between unified VLMs and understanding-only VLMs, trained on understanding data biased on the battery attribute. "_u" denotes understanding-only training. "-D" denotes using separate understanding vision adapter and generation vision adapter.

As shown in Figure 5, when training on data biased on the weather attribute, both understanding-only models (VQ_u and SigLIP_u) struggle to identify weather-related information. In contrast, all unified VLMs trained with mixed data achieve nearly 100% VQA accuracy. Similarly, as shown in Figure 6, for SigLIP-based VLMs, the unified models significantly outperform their understanding-only counterparts. For VQ-based VLMs, although all models exhibit relatively poor performance (likely due to the inherent difficulty of identifying characters with the VQ encoder), the unified VLMs still surpass the understanding-only models by the end of training. These results clearly demonstrate the knowledge transfer from generation tasks to understanding tasks in unified VLMs, partially explaining why increasing generation data can boost understanding accuracy.

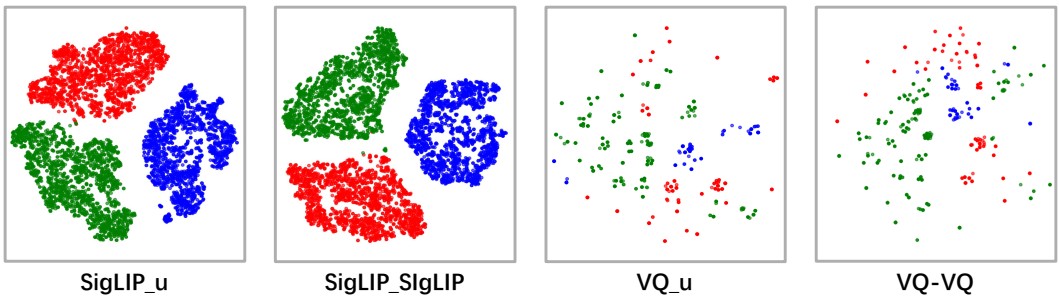

Figure 7: t-SNE visualization of input vision tokens corresponding to the ViT patch representing the weather icon, output by the understanding vision adapter. Samples are colored using the weather ground truth, where red, green, blue refers to sunny, cloudy, and rainy. 5000 samples are shown.

To further investigate how this knowledge transfer occurs, we analyze the understanding vision adapter. We hypothesize that generation training may force the vision input and output spaces to retain more image-related information. However, we conclude that this is not the primary factor driving the observed knowledge transfer.

We analyze the understanding vision tokens output by the understanding vision adapter in SigLIP_u, SigLIP-SigLIP, VQ_u, VQ-VQ trained on the weather-biased dataset. For each model we sample 5000 vision tokens corresponding to the ViT patch representing the weather icon in 5000 images respectively. Using t-SNE, we visualize these tokens colored according to the weather ground truth, as shown in Figure 7. The results reveal no confusion in weather labels across all models. Additionally, we perform linear probing for the vision tokens from each model on a training/testing split of 4K/1K samples. After 10 epochs, vision tokens from all the four models, including the understanding-only models, achieve 100% linear probing accuracy. This indicates that weather-related information is present in the vision tokens of both unified and understanding-only VLMs.

The above analysis indicates that knowledge transfer is not primarily driven by generation training forcing the vision input space to retain more information. Weather-related information exists in both unified and understanding-only VLMs. However, in understanding-only training, the base LLM

fails to utilize this information, neglecting the relationship between vision tokens and text tokens. In contrast, with generation training, the base LLM learns this relationship well. This suggests that the base LLM is capable of implicitly aligning the vision input and output spaces to some extent, allowing the relationships learned in the vision output space to generalize to the vision input space.

To further validate our findings, we experiment with separate understanding and generation vision adapters instead of sharing their parameters (default setting). As shown by **SigLIP-D** (D for "detach") and **VQ-VQ-D** in Figures 5 and 6, we observe that parameter sharing between the two vision adapters is not the key factor enabling knowledge transfer in unified VLMs, which also leads to the explanation that the base LLM is capable of implicitly aligning the vision input and output spaces.

### 4.3 Applications on Real-Case VLMs

To further validate our findings in real-world scenarios, we conduct experiments based on LLaVA-V1.5-7B Liu et al. (2023). We extend LLaVA by adding a generation vision adapter and a generation vision head, enabling it to generate image CLIP embeddings (Tong et al., 2024) or VQ-VAE token IDs Chen et al. (2025). For VQ-VAE we use `vq_ds16_t2i`, same with the previous experiments, but use a generation vision adapter of 2-layer MLP instead of a single linear layer, since the data is more complexed. For the image generation data, we reverse the image-caption pairs from ShareGPT4V (Chen et al., 2023). Specifically, we sample 350K image-caption pairs from the 1.2M ShareGPT4V-PT dataset, using the image captions as generation instructions and the corresponding images as generation targets.

Following the two-stage training process of the original LLaVA framework, we proceed as follows: In the first stage, we freeze the base LLM and vision encoder while updating the understanding vision adapter, generation vision adapter, and generation head. To this end, we augment the original 558K image understanding dataset with an additional 150K image generation samples. In the second stage, we unfreeze the vision encoder and update all other parameters. Here, we further augment the original 665K instruction-tuning dataset with 200K image generation samples.

Table 2: Performance comparison between original LLaVA-1.5-7B (understanding-only) and the unified VLM version with 350K additional image generation data (150K for pre-training stage and 200K for instruction-tuning stage). Results for the original LLaVA-1.5-7B are from the official report of LMMs-Eval Li et al..

| Model | MME | MMBench_EN | POPE | VizWiz | MMVet | GQA | MMStar | TextVQA |
|---|---|---|---|---|---|---|---|---|
| LLaVA | **1510.7** | 64.3 | 85.9 | 54.4 | 30.6 | 62.0 | 33.3 | 45.8 |
| CLIP-CLIP | 1506.4 | 65.0 | 86.5 | 55.4 | **34.4** | 62.0 | 36.2 | 47.0 |
| CLIP-VQ | 1476.6 | **66.1** | **86.6** | **58.1** | 31.2 | **62.1** | **36.3** | **47.1** |

We evaluate the MLLMs on eight popular independent MLLM benchmarks (Hudson & Manning, 2019; Liu et al., 2025; Fu et al., 2023; Chen et al., 2024a; Yu et al., 2023; Li et al., 2023; Gurari et al., 2018; Singh et al., 2019). As evaluation results shown in Table 2, incorporating generation tasks during training does not conflict with understanding tasks. The unified version achieves non-trivial improvements on most benchmarks compared to the understanding-only version. This further strengthens our findings and demonstrates the potential of unified VLMs with mixed training of understanding and generation.

### 5 Conclusion

This work systematically investigates the generalization across understanding and generation in various unified vision-language models (VLMs),. Our findings reveal three key insights: **(1)**: Mutual benefits between understanding and generation exist across multiple unified VLMs. **(2)**: The alignment between vision input and output spaces is a critical factor for cross-task generalization. Better alignment will lead to better cross-task generalization. **(3)**: Knowledge learned in generation tasks can transfer to understanding tasks, even when there are gaps between vision input and output spaces. These results validate the hypothesis that unification fosters synergies between understanding and generation, and underscore the necessity of unified VLMs, offering actionable guidelines and insights for model design.

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

## A    APPENDIX

## B    EVALUATION PROCEDURE FOR SMARTWATCH

We compute the VQA accuracy score across four key attributes: time, weather, position, and battery. For weather and position, we use matching-based evaluation. For time and battery, given their more continuous value ranges and the model's progressive learning nature, simple matching-based evaluation cannot fully capture the model's ability. Thus, we compute time accuracy as $1 - \frac{error_h}{6} - \frac{error_m}{30} - \frac{error_s}{30}$, and battery accuracy as $1 - \frac{error_b}{100}$.

## C    ADDITIONAL EXPERIMENT DETAILS

For all experiments conducted on the synthetic SmartWatch UI dataset, we utilized 3 Nvidia H100 80GB GPUs. Training was performed for one epoch with a global batch size of 393. To optimize GPU memory usage, we employed Low-Rank Adaptation (LoRA) (Hu et al., 2022), setting `lora_r` to 128 and `lora_alpha` to 256, with a learning rate of $2e - 4$ following the configuration in LLaVA-1.5. The learning rates for the understanding vision adapter, generation vision adapter, and image generation head were set to $1e - 4$. For the default settings of 60K generation data and 120K understanding data, the entire training process completed within an hour. Additionally, we independently generated 1.5K samples as the test dataset, with 1,000 samples for Visual Question Answering (VQA) and 500 samples for text-to-image generation.

For the final real-world case experiment, we strictly followed the implementation details of LLaVA-1.5, using all default hyperparameters. We do not use LoRA in this experiment. We used 8 Nvidia H100 80GB GPUs. The pre-training stage completed within 3 hours, while the fine-tuning stage finished within 6 hours.

For all unified VLMs that generate CLIP or SigLIP embeddings, we applied cosine similarity loss on the generation part, following (Tong et al., 2024). For unified VLMs that generate VQ-VAE token IDs, we used cross-entropy loss. In the case of SigLIP-VQ unified VLMs, considering the scale difference between the cross-entropy loss of VQ-VAE token IDs and language token IDs, we set the weight of the generation loss to 0.2.

## D    LIMITATIONS

First, our study does not include unified VLM architectures that incorporate diffusion-based generation components, such as Emu3 (Wang et al., 2024b) and Transfusion (Zhou et al., 2024). Second, while the mutual benefits between understanding and generation tasks are clearly demonstrated in our experiments using the current LLM base, adopting a more advanced LLM base could potentially yield even more promising results.

## E    DETAILED EXPERIMENT RESULTS

Table A3: Performance comparison of various unified VLMs with understanding-only or generation-only VLMs. "_u" refers to understanding-only; "_g" refers to generation-only; "*" refers to affine transformation after the understanding vision adapter.

| Model | time_acc | weather_acc | position_acc | battery_acc | total_acc | FID score |
|---|---|---|---|---|---|---|
| SigLIP_u | 49.7 | 100.0 | 93.4 | 56.3 | 76.6 | - |
| SigLIP-VQ | 66.5 | 100.0 | 100.0 | 69.2 | 85.6 | 198.2 |
| SigLIP-SigLIP | 95.8 | 100.0 | 100.0 | 98.4 | 98.7 | - |
| VQ_u | 50.8 | 100.0 | 100.0 | 61.6 | 80.5 | - |
| VQ_g | - | - | - | - | - | 210.9 |
| VQ-SigLIP | 45.8 | 100.0 | 100.0 | 81.2 | 85.5 | - |
| VQ-VQ | 52.7 | 100.0 | 100.0 | 96.8 | 89.3 | 190.2 |
| SigLIP*_u | 50.4 | 100.0 | 98.8 | 49.5 | 78.2 | - |
| SigLIP*-SigLIP | 54.3 | 100.0 | 100.0 | 98.9 | 91.3 | - |
| VQ*_u | 52.4 | 100.0 | 97.9 | 56.9 | 78.9 | - |
| VQ*-VQ | 60.0 | 100.0 | 100.0 | 80.6 | 85.1 | 195.8 |

Table A4: Performance of different unified VLMs along the increase of understanding or generation data.

| Model | Und_data | Gen_data | time_acc | weather_acc | position_acc | battery_acc | total_acc | FID score |
|---|---|---|---|---|---|---|---|---|
| VQ-VQ | 0K | 60K | - | - | - | - | - | 210.9 |
| VQ-VQ | 120K | 60K | 52.7 | 100.0 | 100.0 | 96.8 | 89.3 | 192.2 |
| VQ-VQ | 180K | 60K | 56.0 | 100.0 | 100.0 | 97.5 | 90.2 | 188.1 |
| VQ-VQ | 240K | 60K | 73.7 | 100.0 | 100.0 | 97.9 | 94.0 | 165.1 |
| VQ-VQ | 300K | 60K | 62.8 | 100.0 | 100.0 | 98.4 | 91.8 | 152.0 |
| VQ-VQ | 120K | 0K | 50.8 | 100.0 | 100.0 | 61.6 | 80.5 | - |
| VQ-VQ | 120K | 60K | 52.7 | 100.0 | 100.0 | 96.8 | 89.3 | 192.2 |
| VQ-VQ | 120K | 90K | 48.3 | 100.0 | 100.0 | 96.2 | 88.3 | 180.8 |
| VQ-VQ | 120K | 120K | 48.3 | 100.0 | 100.0 | 80.7 | 84.5 | 171.4 |
| VQ-VQ | 120K | 180K | 60.0 | 100.0 | 100.0 | 98.1 | 89.3 | 116.4 |
| SigLIP-VQ | 0K | 60K | - | - | - | - | - | 210.9 |
| SigLIP-VQ | 120K | 60K | 66.5 | 100.0 | 100.0 | 69.2 | 85.6 | 198.2 |
| SigLIP-VQ | 180K | 60K | 80.4 | 100.0 | 100.0 | 100.0 | 95.9 | 183.0 |
| SigLIP-VQ | 240K | 60K | 54.9 | 100.0 | 100.0 | 100.0 | 90.6 | 162.4 |
| SigLIP-VQ | 300K | 60K | 97.0 | 100.0 | 100.0 | 100.0 | 99.3 | 157.0 |
| SigLIP-VQ | 120K | 0K | 49.7 | 100.0 | 93.4 | 56.3 | 76.6 | - |
| SigLIP-VQ | 120K | 60K | 66.5 | 100.0 | 100.0 | 69.2 | 85.6 | 198.2 |
| SigLIP-VQ | 120K | 90K | 50.8 | 100.0 | 100.0 | 100.0 | 89.7 | 156.5 |
| SigLIP-VQ | 120K | 120K | 55.0 | 100.0 | 100.0 | 100.0 | 90.6 | 165.6 |
| SigLIP-VQ | 120K | 180K | 96.3 | 100.0 | 100.0 | 100.0 | 99.2 | 131.8 |

Table A5: Performance of different unified VLMs trained on weather-biased dataset.

| Model | time_acc | weather_acc | position_acc | battery_acc | total_acc |
|---|---|---|---|---|---|
| SigLIP_u | 52.9 | 28.6 | 100.0 | 99.7 | 78.7 |
| SigLIP-SigLIP | 81.7 | 100.0 | 100.0 | 97.2 | 95.9 |
| SigLIP-SigLIP-D | 99.6 | 100.0 | 100.0 | 99.9 | 99.9 |
| SigLIP-VQ | 52.2 | 100.0 | 100.0 | 98.6 | 90.8 |
| VQ_u | 50.9 | 54.3 | 81.3 | 77.5 | 70.0 |
| VQ-VQ | 49.1 | 97.1 | 100.0 | 94.4 | 88.7 |
| VQ-VQ-D | 50.7 | 100.0 | 100.0 | 97.4 | 90.3 |
| VQ-SigLIP | 52.8 | 100.0 | 100.0 | 80.6 | 86.6 |

Table A6: Performance of different unified VLMs trained on battery-biased dataset.

| Model | time_acc | weather_acc | position_acc | battery_acc | total_acc |
|---|---|---|---|---|---|
| SigLIP_u | 55.7 | 100.0 | 100.0 | 70.3 | 84.7 |
| SigLIP-SigLIP | 96.6 | 100.0 | 100.0 | 98.0 | 98.9 |
| SigLIP-SigLIP-D | 89.3 | 100.0 | 100.0 | 97.2 | 97.4 |
| SigLIP-VQ | 53.6 | 100.0 | 100.0 | 95.8 | 90.4 |
| VQ_u | 50.1 | 100.0 | 100.0 | 61.3 | 81.5 |
| VQ-VQ | 49.8 | 100.0 | 100.0 | 69.4 | 83.4 |
| VQ-VQ-D | 48.8 | 100.0 | 100.0 | 74.5 | 84.4 |
| VQ-SigLIP | 46.8 | 100.0 | 86.3 | 67.1 | 76.7 |

