# OpenReview forum: "Are Unified Vision-Language Models Necessary: Generalization Across Understanding and Generation"
_ICLR.cc/2026/Conference — Submitted to ICLR 2026_

### Official Review · Reviewer_VtLR · 2025-10-30

**Soundness:** 3
**Presentation:** 3
**Contribution:** 3
**Rating:** 4
**Confidence:** 4

**Summary:**

This paper conducts a systematic study on whether unified vision-language models that integrate both understanding and generation are truly necessary. Using controlled synthetic datasets (SmartWatch and modified CelebA) and several model variants combining SigLIP and VQ-VAE encoders/decoders, the authors show that unified training yields mutual benefits between understanding and generation tasks. They further demonstrate that alignment between visual input and output spaces is crucial—when disrupted, cross-task generalization collapses. Moreover, knowledge learned from generation tasks can transfer to understanding, primarily through the shared base language model rather than modality adapters. Experiments on LLaVA-1.5-7B confirm these findings on real-world benchmarks. Overall, the paper provides compelling empirical evidence and analysis supporting the necessity of unified VLMs and offers design insights for future multimodal systems.

**Strengths:**

1. The paper presents a clear and well-structured experimental setup that allows precise evaluation of cross-task generalization between understanding and generation.
2. The authors provide strong empirical evidence showing that unified training consistently yields mutual benefits across diverse architectures.
3. The analysis of input–output alignment is insightful and convincingly demonstrates how spatial alignment between vision token spaces governs cross-task transfer.
4. The paper offers actionable guidance for future unified VLM design by examining the effects of data scaling and adapter sharing.

**Weaknesses:**

1. The study omits diffusion-based unified architectures, limiting the generality of its conclusions.
2. The evaluation focuses on basic vision-language tasks and does not explore challenging reasoning or compositional generalization cases.
3. The paper does not test SigLIP-based generation in pixel space, as no decoder or visual reconstruction is included, which limits the completeness of the evaluation. An approach is to train a SigLIP-decoder following Emu2 and validate its generation ability.
4. The paper relies on outdated backbones such as Vicuna-7B-v1.5 and LLaVA-1.5-7B, so its conclusions may not generalize to newer multimodal models. Evaluating the proposed findings on recent architectures would better demonstrate their robustness.

**Questions:**

My major concer is in weakness 3 and 4. I'd be happy to increase the score if they are addressed.

---

### Official Review · Reviewer_yW45 · 2025-10-30

**Soundness:** 3
**Presentation:** 3
**Contribution:** 3
**Rating:** 4
**Confidence:** 2

**Summary:**

The authors challenge the common assumption that a unified architecture naturally leads to mutual enhancement between understanding and generation, providing the first systematic, architecture-agnostic study of this phenomenon. Their findings can be summarized as follows. Unified training on both understanding and generation tasks yields mutual benefits, outperforming task-specific models. The alignment of visual input and output spaces is crucial for effective cross-task generalization. Knowledge transfer occurs from generation to understanding tasks, and this transfer is mainly realized within the base LLM, not the modality adapters.

**Strengths:**

1. Good motivation with principled analysis: the paper provides controlled, reproducible experiments to validate the features about unified VLMs.

2. Good experimental design: two datasets with full controllability and detailed ablations (alignment, scaling, bias).

3. Clear practical relevance: In addition to empirical experiments, the paper also provides actionable guidelines (maintaining aligned latent spaces, balancing task ratios).

**Weaknesses:**

1. Limited scope of architectures: The paper excludes diffusion-based or hybrid models (e.g., Transfusion, Emu3). Especially, the generation performance of SigLIP-SigLIP and VQ-SigLIP are missing. In addition, Harmon [a] adopts MAR encoder (different from SigLIP or VQ), which is not discussed and included in experiments. This somewhat narrows the generality of the conclusions.

2. Synthetic bias: Although the synthetic datasets provide control, they are relatively simple compared to real-world multimodal distributions. The gap between these and natural images could limit ecological validity.

3. Lack of deeper theoretical grounding: The study is empirical; a more formal analysis (e.g., information-theoretic or representational similarity argument) could enhance interpretability.

4. Lack of experiments with real-case unified VLMs: The paper presents experiments based on LLaVA-V1.5-7B, but LLaVA-V1.5-7B is not a unified VLM. Why not directly conduct experiments with existing unified VLMs, e.g. Janus?

[a] Wu S, Zhang W, Xu L, et al. Harmonizing visual representations for unified multimodal understanding and generation[J]. arXiv preprint arXiv:2503.21979, 2025.

**Questions:**

1. Can the conclusions of this paper explain the results of previous papers (stated in Line131-142)?

---

### Official Review · Reviewer_2bLg · 2025-10-31

**Soundness:** 1
**Presentation:** 1
**Contribution:** 2
**Rating:** 2
**Confidence:** 3

**Summary:**

This paper examines whether jointly training vision-language models (VLMs) on image understanding and generation tasks leads to mutual benefits. Through controlled experiments with several unified architectures, the authors find that mixed training improves both modalities, especially when visual input and output spaces are well aligned. They also show that knowledge from generation tasks transfers to understanding tasks, supporting the 'necessity' of unified VLMs.

**Strengths:**

- The paper investigates in depth an important and timely topic in multimodal model design. It remains unclear whether specialized or unified VLMs are preferable, and understanding the trade-offs between these approaches is valuable.

- The experimental setup is systematic and controlled, using synthetic datasets that help isolate the factors influencing cross-task generalization.

**Weaknesses:**

- The paper is unclear about how the proposed unified VLMs actually work. Figure 1 doesn’t really help in understanding the architecture, and several components are insufficiently explained. In particular, it’s not clear what the 'generation vision adapter' does: if the LLM outputs image tokens directly, why do these need to be adapted before being fed back into the model? The generation process for the SigLIP-SigLIP and LLaVA settings is also confusing. For models with a VQ decoder, image generation is straightforward, but for those without it’s never explained how the model turns its outputs into actual images (whether through a diffusion module or something else). The lack of generation results for the SigLIP-SigLIP adds to the confusion.

- Although the authors claim to test on 'real-world scenarios', most of the analysis is done on extremely simple datasets of faces and synthetic clocks. Of course, this can be useful to have a controlled setup, but it really limits the practical impact of the findings. It is unclear whether the models trained on these two tasks are of any practical usefulness, or if they just work on these very specific datasets. The choice of models is also extremely limited, as the authors only use Vicuna-7B as a backbone, and even the experiments on LLaVA are restricted to a version finetuned starting from the same base model. It would be important to test whether the evaluated models can handle more realistic multimodal tasks and to verify that the reported findings extend to different LLM backbones.

- The significance of some findings is not entirely clear. For example, the result that most of the cross-task generalization happens within the LLM rather than in the vision adapters seems somewhat expected. Since the LLM has by far the largest number of parameters and is the component where image and text tokens interact through attention, it is natural that most of the transfer would occur there. In Section 4.2, the conclusion that SigLIP encodings already provide semantic visual tokens also feels rather intuitive, so this experiment adds limited new insight. On the contrary, I am unsure whether the same conclusion holds for models using a VQ encoder, as the two panels on the right of Fig. 7 seem to suggest the opposite, and previous studies such as [a] have demonstrated that image tokens in unified models with VQ encoding carry extremely limited semantic content.

**Minor and typos:**
- Line 24: 'this' -> 'these'.
- Line 35: 'has' -> 'have'.
- The orientation of the arrows in Fig. 1 can be confusing.
- Line 50: is 'conversely' the right word? To me, it looks like there is no opposition between the concepts expressed by the two sentences.

**Reference:**

[a] Serra et al., The Narrow Gate: Localized Image-Text Communication in Native Multimodal Models, NeurIPS 2025

**Questions:**

- What is the exact role of the 'generation vision adapter' and why is it required if the LLM already produces image tokens?
- How does image generation work for the SigLIP-SigLIP and LLaVA models?
- Why are the models trained on image captioning data but never tested on such task?
- In the experiment of Fig. 3, are unified and single-task models trained on the same amount of data and for the same number of steps?
- Overall, the paper seems to suggest that unified training is beneficial for understanding and generation performance. However, as the authors report in lines 62-65 real-world unified VLMs do not show a significant edge over single-task models. Do the authors have an explanation for this discrepancy?

---

### Official Review · Reviewer_UE2t · 2025-11-01

**Soundness:** 2
**Presentation:** 4
**Contribution:** 3
**Rating:** 4
**Confidence:** 4

**Summary:**

The paper tests whether putting vision-language understanding (like VQA/captioning) and generation (text-to-image) into a single unified model actually helps, by training on controlled datasets that contain both kinds of supervision, and it finds that unified models do better than task-specific ones on both tasks when they share and align the visual spaces used to read and to generate images; when that alignment is broken, the gains largely disappear, showing that the benefit comes from sharing a common visual representation. Even more interesting, knowledge can flow from the generation side to the understanding side, if the understanding data is missing a concept but the generation data has it, the unified model still learns it, and the authors argue this transfer mainly happens inside the LLM component, which therefore should be the fusion hub in future Janus/Chameleon-like unified VLMs.

**Strengths:**

- Comparison in Fair setting: authors build controlled datasets that contain both understanding and generation signals and then train unified vs task-specific models on the same budget
- Identifying visual-space alignment as the key driver, they break the alignment and show the gains drop.
- The constructed case where generation has a concept and understanding doesn’t, and the unified model still learns it, is a compelling demonstration of cross-task transfer

**Weaknesses:**

- Prior works such as MetaMorph[1] have provided some evidence of mutual benefit transfer between understanding and generation tasks, limiting the novelty of the findings here.
- This work primarily uses smaller datasets built with rule-based text and attribute-style supervision (SmartWatch, templated CelebA) where concepts are annotated, which is not a realistic setting for VLMs.
- The training tasks are closely related on both understanding and generation side: The understanding side is basically attribute VQA/captioning, and the generation side is text-to-image for those same attributes.
- The work draws strong conclusion from narrow evidence


[1] Metamorph: Multimodal understanding and generation via instruction tuning. Tong, Shengbang, David Fan, Jiachen Li, Yunyang Xiong, Xinlei Chen, Koustuv Sinha, Michael Rabbat, Yann LeCun, Saining Xie, and Zhuang Liu. ICCV 2025.

**Questions:**

Experiment with negative-transfer tasks, when one task is noisy or much larger, does it ever hurt the other?

---

### Meta-Review · Area_Chair_ecAP · 2026-01-07

**Summary:**

The authors did not submit rebuttal.

**Reviewer Concerns:**

No concerns are addressed.

**Reviewer Scores:**

Same.

---

### Decision · Program_Chairs · 2026-01-26

Reject